# Factors Involved in the Food Choices of Diners in a Kibbutz Communal Dining Room Buffet: A Qualitative Study

**DOI:** 10.3390/ijerph19031885

**Published:** 2022-02-08

**Authors:** Ofira Katz-Shufan, Tzahit Simon-Tuval, Danit R. Shahar, Paula Feder-Bubis

**Affiliations:** 1Department of Public Health, The Health & Nutrition Innovative International Center, Faculty of Health Sciences, Ben-Gurion University of the Negev, Beer-Sheva 84105, Israel; dshahar@bgu.ac.il; 2Department of Health Policy and Management, Guilford Glazer Faculty of Business and Management and Faculty of Health Sciences, Ben-Gurion University of the Negev, Beer-Sheva 84105, Israel; simont@bgu.ac.il (T.S.-T.); federbub@bgu.ac.il (P.F.-B.)

**Keywords:** diner’s food choices, catering system, dining room buffet, freedom of choice, qualitative research

## Abstract

Improving nutrition improves health outcomes. Eating in a catering system may provide an environment for promoting healthy dietary choices. To map the factors that shape the food choices of diners who routinely eat in catering systems, we collected and analyzed qualitative data about diners’ perceptions of their food choices in communal dining rooms in three kibbutzim in Israel. From May to July 2014, we conducted in-depth, semi-structured, face-to-face interviews with 13 diners who ate at least three lunches per week in the kibbutz’s dining room. Data analysis followed thematic analysis principles. Two categories of themes emerged from the interviews. In the personal context category, the themes identified were eating as a task and attempts to control one’s eating. In the contextual aspects of eating in the catering system category, themes identified were eating in the dining room as a default, the characteristics of the food served, routine, and personal versus public aspects. The sub-theme of the diners’ freedom of choice emerged in the two categories of themes. Diners’ wishes of maintaining their freedom of choice may be an important contribution to the debate of whether catering systems should provide only healthy foods, which may jeopardize diners’ freedom of choice.

## 1. Introduction

The human diet has a significant impact on disease prevention, quality of life, and longevity [1]. Understanding the psychosocial influences on food choices is critical to the development of dietary intervention programs to assist consumers in developing healthy eating habits and promoting a dietary shift toward a healthy diet [2]. Catering systems may provide a promising opportunity to promote a healthier diet for their diners [3,4,5], especially through the creation of a healthy food environment. The factors known as influencing food choices include food-related internal factors (sensory and perceptual features), food-related external (environmental) factors, personal factors (physiological and psychological characteristics, habits and experiences), cognitive factors (knowledge, attitude and personal identity) as well as sociocultural factors [6]. Environmental changes in catering systems have been shown to have a small-to-medium effect on improving diners’ nutritional choices [7]. Understanding the factors involved in the food choices of diners in catering systems may increase their responsiveness to nutritional intervention programs [5].

The Nutrition Environmental Kibbutzim Study (NEKST) [8] is an interventional study designed to evaluate the effect of an integrated environmental intervention program on regular diners’ food choices in communal catering systems in Israeli kibbutzim. A kibbutz is a uniquely Israeli way of living, a collective rural community with several hundred members [9]. It is characterized by a socioeconomic system based on principles of equality, joint ownership of property, and cooperation in production. In line with these principles, the kibbutz has a communal catering system. Eating in the kibbutz dining room is a routine part of kibbutz members’ daily lives. Kibbutz members and their families have their meals together. Thus, apart from being the food provider, the kibbutz dining room also serves as a meeting place for its members. Furthermore, some of the kibbutz members are part of their kibbutz’s kitchen and dining room staff. Kibbutz members and their families may have their meals together, yet they can also choose whether and how to use this communal facility. Alternatively, they can eat at home using food products available in the market. There is usually an industrial area with factories around a kibbutz. The workers in these factories often eat their lunches in the kibbutz dining room. The kibbutz catering system’s buffet style meal for lunch has a variety of dishes (eight main dishes—rich in protein, three carbohydrate side dishes, 7–10 vegetable dishes, one legume dish and two kinds of soup) that repeats itself regularly and is familiar to the regular diners. The nutritional benefits of the food in the kibbutz dining room are the accessible, fresh, and diverse cooked food served. Conversely, no healthy food policy is routinely implemented in these communal dining rooms. In most cases, they do not consult with a dietitian regularly to cook and serve healthy food. Therefore, the dishes served often include extra-processed food products and the use of food ingredients rich in undesired nutrients such as sugar, salt, saturated fat and trans fats. The payment for eating in this catering system is charged to the account of each kibbutz member. The owners of the factories subsidize the meals for their workers.

The aim of this study was to utilize this unique setting to provide a more nuanced understanding of the factors involved in the food choices of regular diners in a kibbutz communal catering buffet. To achieve this goal, we adopted a qualitative approach, for the examination of the perceptions of the diners about food, their food choices and healthy nutrition in the context of eating in the kibbutz dining room. 

## 2. Methods

A qualitative research design was undertaken in order to elicit the diners’ perspectives. The study took place from May to July 2014 at three kibbutzim located in the south of Israel. 

### 2.1. Recruitment

Members of the three kibbutzim were asked via an advertisement to voluntarily participate in an interview. The advertisement was sent by email to the kibbutz community and was posted on bulletin boards in the dining rooms. The advertisement requested the participation of regular diners who ate at least three lunches a week in the communal dining room. First, we recruited 10 kibbutz members who responded to the advertisement. As is customary with qualitative studies, this initial convenience sampling [10] was followed by criteria sampling [10] in which we reached out to participants for whom healthy nutrition was a less important value. The head of the health committee of each kibbutz, who served as the contact person for the study, facilitated this step. In this second stage, we recruited three participants. Interviews were conducted until theoretical saturation, meaning no new information emerged from the interviews or was reached [11].

### 2.2. Data Collection

Thirteen diners who ate at least three lunches per week in a kibbutz dining room were interviewed using in-depth, face-to-face interviews. A semi-structured interview guide was developed, one that allowed each individual to tell his or her own story [12]. The relational approach [13] informed the writing of the interview guide. This approach was based on the first author’s (OKS) personal experience as a casual diner in a kibbutz and as a consultant dietitian in similar dining settings, as well as her acquaintance with kibbutz dining rooms and diners. The interview guide included mostly open-ended questions to obtain rich, detailed information from each participant [14].

The topics addressed in the interview are presented in Table 1. These topics included perceptions about healthy eating, the diners’ food choices in the dining room, and opinions about the responsibility of healthy eating in the dining room. After the first three interviews, we reviewed the interview guide. In light of the participants’ ease when answering the questions and the comprehensiveness of their responses, no changes were made to the guide. The interviews were conducted in Hebrew by the first author (O.K.-S.), a registered dietician and a member of one of the three kibbutzim in this study. The interviews took place considering each participant convenience. After providing an explanation about the purpose of the interview, the interviewer recorded the participants’ verbal consent to participate in the study. The interviewer explained to each participant that s/he might opt out questions or withdraw from the interview at any point in time. All interviews were audio-recorded and fully transcribed. 

### 2.3. Data Analysis

The first author (O.K.-S.) conducted the data analysis using thematic analysis principles [15]. The steps of the data analysis process were familiarization with all of the texts, identifying the thematic framework, applying it to all of the data, mapping and interpreting the meanings, and finding associations between themes [16]. The development of the themes was based on an abductive approach [17] that combined insights from the data collected (an inductive approach) and from the investigator’s prior theoretical understanding of the phenomenon under study (an a-priori approach) [18]. The thematic framework was further elaborated and jointly discussed with the study team members. The early acquaintance of the interviewees with the main researcher (O.K.-S.) as a dietitian, whose intention to plan a program to improve nutrition in the dining room was considered during data collection and analysis. Given the interviewer’s personal involvement in the research field, self-reflection was part of the data analysis to ensure the transparency, rigor and credibility of the results [19]. Both the interviewer and participants were native Hebrew speakers; thus, the quotes we cite were translated into English. Following the recommendations of cross-language qualitative research, the material was translated into English after we completed the thematic analysis and after the selection of representative quotes. In addition, a professional English translator was consulted to ensure that the translation of unique Hebrew expressions and phrasing retained the original meaning in context [20]. IBM SPSS Statistics for Windows (version 24.0, Armonk, NY, USA: IBM Corp.) was used for descriptive analyses of the interviewees’ personal characteristics.

### 2.4. Ethical Approval

Ethical approval of the study was granted by the Sub-Committee of Research and Experiments with Human Participation, Faculty of Health Sciences, Ben-Gurion University of the Negev (#2014-15). The names of the kibbutzim and personal details of the interviewees were coded in order to ensure anonymity.

## 3. Results

As presented in Table 2, six of the interviewees were from Kibbutz No. 1, four were from Kibbutz No. 2, and three were from Kibbutz No. 3. Seven were females and six were males. The interviewees’ ages ranged from 22 to 82 years (54.5 ± 19.7), and they had been eating 3–6 lunches per week in the dining room for 4–64 years (35.5 ± 19.9).

The thematic framework categories, themes, and sub-themes that were identified are described in Table 3. We established two categories for the responses: the personal context and the contextual aspects of eating in the kibbutz dining room.

### 3.1. The Personal Context

The category of the personal context involves themes related to the diners’ inner values, thoughts, emotions, and deeds with respect to their food choices and eating habits.

#### 3.1.1. Eating as a Task

Some diners regard eating as a serious, difficult task. They talk about their awareness of the nature of the food and their food choices, and about the difficulty in performing the eating task well, as well as their confusion about it.
*Healthy eating for me is first of all to eat … it’s not something that you can always manage to do, the tasks, first thing to eat according to what the body needs, according to the body and not following your desire, awareness of the essence of food: fresh food, not-processed foods, food ingredients, carbohydrates, legumes, paying attention to that.**(#4)*

This participant feels that she cannot always perform the eating task as she would like, thus ascribing difficulty to it. She underscores the importance of eating according to physical needs rather than emotional ones. She identifies paying attention to the food’s characteristics as an effective mechanism for performing the task of eating.
*Sometimes, I do not know what the dish contains … Sometimes, I do not know if the (food) combinations I make are correct in terms of the composition of the meal, if it fits. Because there are those who say you need protein with it … it always confuses me … So sometimes it’s hard, I don’t always know what’s right, what’s wrong.**(#1)*

Participant #1 expresses confusion about the eating task. This confusion stems from the lack of knowledge about what the food ingredients are, what the adequate food combinations are for her, and the opinions of significant others about the food. Notably, she is sure there is one proper way to choose what to eat. Her perception of the eating task is dichotomous (right or wrong), and she wants a definite answer as to what is right.

***Experience-based eating*** is a sub-theme of *eating as a task*. This sub-theme relates to choosing food based on early acquaintance with the food served and with the dilemmas associated with the task of eating.
*Since I choose, and the choice is already based on experience, so I choose what I like. What I choose, ninety-five percent tastes good to me, even very tasty to me … Even at lunch, I usually stick to some carbs, soup, and a main course and fresh salads, which are always delicious to me.**(#10)*

The considerable experience of these participants allows some of them to choose the food they regard as suitable and delicious. For them, the eating task is clear: choosing what to eat and repeating this choice almost every meal. As experts in the eating task, they know the different foods and dishes, their nutritional content, and their desired meal composition.

Another sub-theme of *eating as a task* is ***eating and emotions***. It addresses both the effect of emotions on food choices and eating and the effect of food choices and eating on one’s feelings and emotions.
*It’s in the emotional, mental, moods part. A very, very central part is the issue of nutrition, which is related to moods in my opinion. I am very interested in healthy eating, also because of trivial health issues for my age, sixty-three, and for my feeling. To me, this issue is very, very important. Health-related issues of nutrition are central to me … Healthy eating begins with eating at more or less regular times. Lots of drinking, especially unsweetened…and diversity between the various nutrients, carbohydrates, proteins, the foods with vitamins…and the foods for the soul, the sweet ones.**(#10)*

Similar to other participants, this participant describes a marked connection between emotions and mood and eating and food choices. Adherence to her healthy eating task, for which she has certain rules, is important to her. She also describes some of the foods as food for the soul, meaning that she eats them for emotional reasons, not nutritional ones.
*Food is a very, very, very serious issue, and it definitely affects mood, and emotion. You are more satiated, happier, I ate, I had fun, it was very pleasant. Very meaningful… you are more satisfied, you are calmer, you work more quietly….**(#3)*

Again, the task of eating is regarded as a serious issue. This task affects both mood and emotions. The eating itself brings emotional improvement. Conversely, the emotions also affect eating and food choices:
*Sometimes you…feel less good, or you are nervous, or you are busy and things like that, sometimes you take more than you need.**(#1)*

For Participant #1, contingencies and emotions affect her food choices, specifically the amount of food she chooses.

The third sub-theme of *eating as a task* is ***eating to satisfy physical needs***. This sub-theme includes satisfying physical hunger and hedonic hunger (i.e., eating for pleasure).

When asked about what he chooses to eat in general, Participant #7 says:
*First of all, that the food will be tasty to me. Everything else is less important to me, and that the portion will be large enough. Usually this is not a problem because the portions are quite standard.**(#7)*

For this participant, the most important factor in the eating task is eating according to what tastes good. The second most important factor is to alleviate hunger, meaning to choose a sufficient amount of food. By saying “*Everything else is less important to me,*” he implies that healthy eating and nutrition are less important for him.
*First thing, I check with myself. I already know if I come (to the dining room) hungry or not. Sometimes, I do not come really hungry, so it is easier to choose … I think about whether it is healthy or not, is it fattening, if it has a lot of oil in it … if it satisfies me.**(#1)*

In order to accomplish the eating task, a self-check seems to be useful to become aware of the immediate needs of one’s body such as hunger. Later on, the participant debates whether what she chose was satisfying, had the right nutritional value, or was suitable for weight control.

***Eating according to the diner’s values*** is another sub-theme of the eating task. One of the values that emerges from the interviews is the ***freedom of choice***.
*I choose what I want … meat, a lot of … meat. I do not eat only meat, also carbohydrates, vegetables, and all sorts of things, but I love meat. If you ask about my preferences … also chicken, fish, fish less, chicken, and beef.**(#7)*

This diner expresses the value of freedom of choice by stating that he eats only according to his preferences. His words “*but I love meat*” imply his freedom of choice: this is what he wants, not something else, and he wants to be free to choose this food, although meat might be considered an unhealthy food. He then adds, “*If you ask about my preferences.*” This phrase implies that he is surprised by the opportunity to express personal preferences regarding foods that are not regarded as healthy, even after being informed that the interviewer is a dietitian, and the study is about improving nutrition.

Additionally, for several diners, ***healthy eating as a value*** plays a fundamental role in the eating task. Diners express clear ideas about what healthy eating is and exhibit a particularly dichotomous view about “good” and “bad” foods. Furthermore, some diners regard healthy food as not tasty.
*Healthy eating is, first and foremost, having a regular eating pattern, eating in moderation … in moderation in every way. Both the amounts we put in our mouths and the nutritional values we put in our mouths, it goes in both directions, like, neither too much nor too little. Maybe, mostly with thought about what we put into the body. Of course, with the knowledge that we know about what healthy food is, what harmful food is… choosing foods that we know are healthier and that we know are less harmful to the body I would say … eating vegetables, oily foods are very fatty and harmful … I think everything related to margarine is harmful. Everything that is deep-fried in oil, like French fries or schnitzel … In general, everything that is fat, trans-fat, is harmful, and those are present in almost all processed foods. If we are talking about lunches in the dining room, it is the processed schnitzel and pastry with margarine, things like that.**(#9)*

This participant’s words about what, in what way, and how much is healthy to eat highlight ***healthy eating as a value*** and as a part of the task of eating. This diner knows about healthy nutrition and can explain what healthy nutrition is. He divides food into “good” and “bad” by, for example, stating that vegetables are “good” food, and oil, oily foods, fried foods, and processed foods are “bad” foods.
*When I see that food is not aesthetic and that it is floating in oil, I do not approach it at all … Aesthetics speaks very much to me. This unhealthy fat, all the oils there … clog the arteries completely.**(#3)*

In Participant #3′s words, “bad” foods are also described as unaesthetic. Thus, they should be avoided in order to preserve healthy eating as a value. Here again, as in the dichotomous thinking described earlier, oily food is described as the worst food. The participant regards the sight of it as unpleasant. Its appearance and “bad” nutritional values are the reason to avoid them in order to improve health.
*It is important to me that the food is delicious. Sometimes, it’s hard because not always the healthy things are the tastiest, but I have found the healthy things I love. It’s a matter of tasting things, say quinoa, they were eating at my house all the time and I was just now, just on the last diet, I started eating quinoa. And now, it’s really tasty for me.**(#1)*

This diner describes the conflict in choosing between delicious food and healthy food due to the perception that healthy food is less tasty. Moreover, this perception makes foods that are linked to a diet even difficult to taste.

#### 3.1.2. Attempts to Control One’s Eating

The participants who want to engage in the eating task in what they regard as the optimal way may try to control what they eat. Doing so involves an internal discussion about controlling their eating. Attempts to control their eating as part of a weight loss diet influences the diners’ thoughts, emotions, and food choices.
*Usually even if there are bread rolls that look good, I will take a slice of bread … not more than a slice. On the way to the table, I ask myself if maybe I would give up the bread…I come in thinking: “Today I will only eat vegetables and salad.” Lately, I feel I need to lose some weight.**(#4)*

Diners stipulate and follow rules for weight loss. In the case of Participant #4, she will rarely eat bread, and if she does, she limits its amount. She has internalized these rules such that they guide her to choose foods that fit her diet, even if she dislikes the taste. Another dichotomy is evident here—that eating only “good” food such as vegetables will lead to weight loss. Another dichotomous division emerges between dieting and non-dieting periods. In addition to the division between “bad” food and “good” food, the behavior of diners is different during these two periods:
*In the period I keep my diet, I define it as a diet period, I have a preference for fish. But only if the fish is really suitable, and not immersed in oil. Even in the choice of main courses, from a health perception, I avoid beef even though I really like it.**(#10)*

In a dieting period, when following the diet’s rules, the diners try to choose the foods that fit the diet’s rules and avoid foods that they consider tasty.

This internal discussion includes conflicts that involve ***morality, guilt, and remorse***. Morality is represented by the thoughts about what is good and what is bad in a dichotomous way. Guilt is the emotion that arises from this moral conflict in choosing between what is perceived as good or bad. Remorse is the emotion that arises following the choice of what is regarded as bad. This whole emotional process is described as accompanied by bad feelings and regret.
*If there is too much floating oil, I will not choose this dish. I avoid eating red meat, not that I dislike it, but if I see that I have a choice between chicken, turkey, or beef, then I take either turkey or chicken. I try…due to nutritional considerations. I just think it’s better for me to eat…to reduce the amount of red meat … I prefer to take more vegetables, even if the vegetables are cooked. There are things I do not love, so I will not take them. I prefer to take salads that are without mayonnaise … I avoid desserts for lunch, for example. I will leave this privilege until the afternoon … to reduce the amount of sugar and carbohydrates.**(#8)*

Participant #8 assesses the ***morality*** of his food choices using dichotomous perceptions about “good” foods such as vegetables or “bad” foods such as red meat. Moreover, he sees eating sweet desserts for lunch as unsuitable; therefore, choosing them may become a moral dilemma.
*I was debating whether to eat the chicken boiled in water … or the chicken I chose. I try to make it mostly chicken boiled in water. I feel like this week, which is a week I did not meet my diet rules… so I say to myself: I’ve already ruined this week’s diet, so I kind of give up and go for something tasty.**(#9)*

This internal discussion continues and deepens. It begins with the matter of morality: a chicken cooked in water is “good” food, and eating only good foods follows the rules the diner has set for himself. Non-compliance with these self-imposed rules leads to guilty feelings that, in turn, lead him to break his own rules. He is trapped within his own rules. Breaking them repeatedly results in guilt and remorse, creating a vicious circle.
*Borekas (a filled pastry)—if I eat them, I know I’m committing a crime. Puff pastry is a kind of poison. I do not eat large quantities of it. I am aware that it is processed food, but from time to time, it seems fine to me… Willpower, willpower… It was borekas, which I don’t think have any nutritional value that can add anything to me. Sometimes, there is a dish that looks great, for example, creamy pasta, and I hold myself from taking it because it is not something I need.**(#4)*

This participant describes attempts to control her eating that are accompanied by strong feelings of guilt. She has a dichotomous perception of food as good or bad, characterizing the latter as “*poison*”. She tries to assert her willpower to control her eating, which implies that great strength is needed to control eating. In her words, she minimizes her actions by saying that she eats only a little of these “bad” foods. This expression may indicate the feelings of guilt that accompany her actions.
*I try to pick my food so that it makes more sense. I break down almost every night … (trying) without all the snacks and without all the breads. Bread is one of my biggest loves. I do not succeed. Bread breaks me (Bread is my weakness)—It kills me.**(#3)*

In the attempts to control her eating, Participant #3 tries to use logic, but does not succeed, and is remorseful about her choices. Her great regret is implied in the personified descriptions of food such as, “*Bread is one of my biggest loves*”.

### 3.2. The Contextual Aspects of Eating in the Catering System Category

This category includes themes about the interrelationship between the diner and the specific catering system. It includes the setting, the food served, the role of the catering system in the diner’s routine and its social context.

#### 3.2.1. Eating in the Dining Room as a Default

The kibbutz dining room is available and accessible to all diners. Nevertheless, few interviewees mentioned the opportunity, accessibility, and privilege of eating daily in a place that offered a wide variety of hot and fresh food. Diners did not talk about this point, maybe because eating in the dining room is a taken-for-granted option in the kibbutz. The diners mentioned the benefits of eating in the dining room as ***convenient and suitable for the diner’s needs*** but did not describe it as an opportunity nor an advantage. Moreover, they noted the disadvantages of eating in the dining room, as it ***does not suit all diners***. The diners talked about the difference between eating at home and eating in the dining room—especially about preparing food at home, which was described as a burden, but also as preferable to the food in the dining room. Some diners mentioned convenience as a reason they chose to eat in the dining room. Others thought that eating at home offered the ability to choose food that was more to one’s personal tastes or nutritional goals.
*I’m really looking at the options I have. I’m not one that will cook a dish for myself for tomorrow, okay? I do not have the time, I would not do it. This is the best I can get right now. I’m sure that if I could eat at home, then I would be preparing myself a healthier meal. But this is the situation at the moment, so it seems to me that it is better to eat in the dining room.**(#13)*

Similar to other diners, Participant #13 prefers home-cooked food to the dining room food, although, for reasons of convenience, he chooses to eat in the dining room.
*Usually, I’m quite happy with our dining room. In my opinion, it functions quite well. There is a pretty large selection of dishes. I have an occasional criticism here and there, but overall, I think it more or less meets my needs. The food in the dining room usually satisfies me. It’s my lunch almost regularly … it’s the food composition I need, and it meets my health needs.**(#5)*

As a regular diner in the dining room, Participant #5, who is deeply familiar with it, has strong opinions about the food served and how the catering system functions, for better or worse. For him, the regular habit of eating in the dining room and the food served there seems appropriate, prompting him to continue to choose to eat there. Some diners find eating in the dining room to be a compromise, feeling that it does not meet their needs.
*Because there are things that I already really know how they are, what they taste like, I do not approach them at all. All sorts of things they do not know how to prepare, like whole rice. I am eager to eat it, but it is like stones. So many times I have taken it and added sauce that I know is unhealthy, so I say well, okay, but I know I did not do well. My awareness of choosing food in the dining room is very high, and it is very difficult for me to eat there.**(#3)*

Participant #3 also seems to have a deep familiarity with eating in the dining room. She has strong opinions about the food served. Despite her many efforts to eat in the dining room, the food there does not suit her. She is disappointed and criticizes the characteristics of the food served as unappealing or unhealthy.

#### 3.2.2. The Characteristics of the Food Served

This theme involves the availability, diversity, appearance, flavor, freshness, health (nutritional components) and price of the food. For example, Participant #4 says:
*I eat every day in the dining room … it happens to me that I take food and tell myself you are putting nonsense into your body. I would choose something else if I had another alternative. There are situations like this, there are not many.**(#4)*

Participant #4, an experienced eater in the dining room, considers some of the food served as “*nonsense*” (un-wanted food), meaning that it does not meet her definition of healthy food. She also criticizes the unavailability of other types of dishes that could help her choose better.
*It should be aesthetic and have less fat. There are salads that have long since prepared. There are the same salads all week…the food should be more delicious, and not at crazy prices that do not fit our budget.**(#3)*

Participant #3 talks about the appearance of the food, its nutritional composition, especially the fat content, and asks for diversity, freshness, and reasonable prices.

#### 3.2.3. Routine

This theme involves eating in the dining room on weekdays as a part of the participants’ workday routine, differentiating it from eating on weekends and holidays.
*There were fries that everyone pounced on, and I did not. I did not even consider it. Even though it was a holiday yesterday, fries are not something I even consider. Yet, I still eat, so-called making a sin occasionally. But it almost never happens at lunches at work … because I am at work. I am in some routine. I put myself in some routine that there is food that is out of bounds. It’s because I force myself… That is also the law, one of the unwritten laws … Friday is the weekend, and I’ll take more. In my defense, if there were not all these rules that I have, then I would eat this delicious food, I would not eat cooked vegetables.**(#9)*

During the week, as part of his routine in his eating task, this diner uses one set of rules to help him control his eating. On weekends and holidays, he has a different set of rules that are less restrictive. This self-imposed routine provides him with what he considers a healthy eating framework.

#### 3.2.4. Personal Versus Public Aspects

This theme includes the interrelationships between the catering system and the diners, and their effects on each other. Many of the participants talked about issues of personal concern as opposed to public ones. The sub-theme of ***responsibility for healthy eating*** is clearly a complex issue. Participants responded to this question intuitively, but continued to debate about the level of responsibility of the diner and the caterer. The interviewees had an on-going inner discussion, and an external discussion between themselves and the catering system, about the responsibility for healthy eating.
*The responsibility is only mine. I mean, the dining room can serve a type of food that for me is out of the question. I will not take it. But it is not the responsibility of the dining room. I am responsible for what I eat. I am responsible … There is no doubt that if we think it through, the kitchen of a kibbutz, as in a family, should be more attentive and more responsible for the food, the quality of the food, and the health of the public. But, in the end, the responsibility is on the person himself.**(#5)*

Intuitively, this participant states that the responsibility is solely on the individual. ”Thinking it through”, he has a much broader set of beliefs about what the dining room should be. He introduces an additional worldview in which he sees the catering system as his family that is responsible and committed to his well-being. In the end, he returns to placing the responsibility solely on the individual and avoids criticizing the catering system.
*I think healthy and quality food must be in our consciousness. We have the tools to do it and that would have prevented me from all these dilemmas … I would expect all the main dishes to be home-made dishes. For example, to take out the processed food for reasons of price and health. We have a big kitchen and many workers, so it is possible to cook like at home. I would take out the amounts of oil … I had a world war over it. I do not think the dining room should serve such things. The arguments I had! It’s completely un-necessary … I was upset. They have no mandate to do that. If you think this dish is unhealthy, do not serve it.**(#4)*

As in the previous quote, this participant longs for a warm, family-like atmosphere in the dining room—one that is conscious of the needs of the individual. Unlike Participant #5, she feels that the catering system does not function properly. She feels alienated from the food service. She is angry at it because she expects it to take responsibility for the food served. These two diners feel that they want the catering system to be attentive to them and want to feel that they are partners in the catering system’s decision making. Other participants regard the catering service simply as a technical system that only serves them food.
*I choose from what (the catering) gives me to choose from, so … you can ask me what I think, it’s not that I can really influence … I do not know if it (the responsibility) is 50–50 because maybe you give me all the best things in the world, and I, in the end, I will not take it. So maybe the responsibility is more on them (the catering).**(#6)*

In contrast with the previous two participants who see themselves as having or should have the ability to influence the catering system, Participant #6 accepts that he has no influence on the food served. Thus, he places more responsibility for his food options on the catering system.

Another sub-theme in the larger theme of *personal versus public aspects* is ***freedom of choice***, which is the individual’s opportunity and autonomy to choose from at least two available options, unconstrained by external parties:
*I think the right to choose for an adult is necessary, but in both directions, the choice between healthy dishes and dishes that are perhaps a little less healthy … because the adult is mature and responsible for choosing what he eats.**(#10)*

Participant #10 discusses the freedom of choice in terms of whether the food served will meet the diners’ expectations. Doing so will afford them freedom of choice. It is not enough for her to say that there is the right to choose. She also says, “*But in both directions*”, meaning that it is important for her to emphasize that the choices must be preserved for everyone: for those who are interested in healthy eating and for those who are not. Speaking about “the adult”, she implies that adults reach a point in life where they cannot be educated or change their food choices.

The notion of ***freedom of choice*** is also presented in two different ways. Some diners feel that they personally do not need a nutritional change in the catering system. Making such changes would infringe on their freedom of choice. However, they acknowledge that other diners such as children, adolescents, and sick people do need it. Other diners feel that a nutritional change in the catering system would suit them, but they are afraid that it would hurt the freedom of choice of other diners, who may not be interested in such a change.
*I think you really have to treat people who do not have the consciousness of what is healthy, or they do not treat it that way. For example, even if the (blood) tests are returned with borderline blood glucose levels, then they continue to drink sweetened juice. This is a problem, educating people. I think what you can do is maybe start with the children’s meals. When young children in primary school and youth come to eat, then children will look at these things and try to make good choices. It really won’t affect me.**(#8)*

Similar to the quote from the previous participant, for Participant #8, education designed to change eating habits can succeed only among children and adolescents. Adults, even if they are aware of and in need of a dietary change, cannot be educated.
*I do not think people should be forced to eat food they do not like or, alternatively, be deprived of food they like because it is unhealthy. It is a personal responsibility of people. I do not think the organization itself, which is supposed to provide food and please the majority of the population, should control these things…How would I feel? I would not feel so good about it, but for me, it would not be so terrible. If it were the opposite, then maybe it would have been terrible … If they had only served unhealthy food.**(#9)*

Participant #9 expresses the opinion of those who regard the catering as a system that should satisfy the majority, and therefore provide everyone with what they like. In this case, freedom of choice would be maintained. Here, he expresses the fear that a change would hurt the freedom of choice for other diners, but not for himself.

## 4. Discussion

The institution of the kibbutz communal dining room provides an excellent opportunity to examine factors that underline diners’ food choices, characterized by a rich selection of dishes. Two main categories of themes emerged as shaping these choices: personal factors and those related to the context of eating in the catering system. In the personal context category, the themes identified were eating as a task and attempts to control one’s eating. In the contextual aspects of eating in the catering system category, the themes identified were eating in the dining room as a default, the characteristics of the food served, routine, and personal versus public aspects of eating in a catering system. The themes and sub-themes in those two categories are closely associated with the known factors that influence people food choices in general [1]. Surprisingly, diners’ freedom of choice arose in both categories, underscoring the importance of the issue among diners and are therefore important when planning an intervention program for catering systems.

Health and weight control are factors that are usually considered in the food choice process [21]. Paradoxically, individuals who are concerned with food intake in relation to health and weight control may be particularly susceptible to overeating and to unsuccessful attempts to control their eating [22]. Diners in our study for whom health was an important value and those who were concerned about weight loss tried to properly address the eating task. They did so through attempts to control their eating. In those attempts, dichotomous thoughts such as the perception of foods as “good” or “bad” or the distinction between routine consumption versus that on weekends and holidays led diners to a moral conflict regarding their eating. In many cases, they felt guilt and remorse when their attempts to control their eating failed. The vicious cycle of the sense of a moral obligation to eat “properly,” followed by the guilt and remorse at the failure to do so, actually undermined their ability to control their eating, with similarity to previous knowledge [23].

Regarding the contextual aspects of eating in the catering system, one theme dealt with the issue of who is responsible for the diners’ healthy eating. Food providers may offer unhealthy foods, which are often desired the most [21], thus increasing the diners’ satisfaction and the profitability of the catering system. Conversely, there may be increased expectations that caterers serve healthy foods [24], although it may be challenging for the catering system [5]. Participants regarded the responsibility for eating healthy as shared by the diners and the catering system. Some of our participants wanted to be and even sometimes felt part of the catering system. They wanted to have the ability to influence the food served. Others felt they had no influence on it. Folk et al. [25], in their study of other kibbutz communal catering systems, suggested that diners should be involved in the decision-making process of nutritional change in the catering system to facilitate effective intervention programs. In contrast to our results, interviewees in Pridgeon et al.’s [26] study, who were diners and part of the catering staff, felt that no such shared responsibility existed. They regarded the diners as responsible for their own choices and rarely saw the catering system as such. This difference may stem from the fact that most of our participants were members of the kibbutz in which the catering system was a community facility that served them and their families. As such, it served a function beyond only a catering system setting. Moreover, historically, some of those diners had been part of the staff of this catering system (as part of their community duties), and most of them had been eating in this setting for many years. Thus, the issue of responsibility might be more complicated and more confusing for this unique group of diners, compared to workplace catering services that are often outsourced. In such situations, employees are not and do not expect to be part of the decision about the food served.

Freedom of choice emerged as a theme in both main categories, hinting it is central in the context of food choices in general and food choices in a catering system in particular. Diners expressed concern about the possible violation of their freedom of choice and their desire to preserve it. They regarded the autonomy to choose from various foods as an ethical discussion involving personal preferences versus public policy [27]. This finding is instructive, given the current debate about whether a top-down, public health approach should be adopted that provides only certain foods or whether all types of foods, including unhealthy ones, should be offered, allowing people to choose based on their preferences, needs, and personal values. Individuals use their capacity for autonomy to express preferences regarding food choices. The challenge is to integrate food choice architecture for healthy eating with diners’ freedom of choice [28]. Our findings show that when planning policies aimed to influence diners in the catering system to choose healthier foods, the policy should take into consideration the diners’ wishes for freedom of choice.

The limitations of this study include the fact that it was conducted in a unique setting and among the unique population of kibbutz members in Israel. This unique setting makes it difficult to apply the results to other populations. However, the results may be useful for other contexts in which the diners are involved in the decisions of the food served by the catering system. Second, although we tried to recruit participants who would express a wide range of opinions, the sampling approach may have resulted in the inclusion of diners who were interested in health and nutrition. Third, the first author (O.K.-S.) is a member of one of the kibbutzim in the study, a fact that could bias the results and data analysis. For this reason, she engaged in self-reflection and periodic discussion with other members of the study team as part of the data analysis.

## 5. Conclusions

This work identified the factors involved in the food choices of those who eat regularly in a catering system in a kibbutz. The two categories of themes found dealt with the personal context and the contextual aspects of eating in the catering system. In the personal context category, experience, emotions, physical needs, values, and perceptions were found as factors of eating as a task and of attempts to control one’s eating. In the context of eating in the catering system category, food-related internal and external characteristics were found as factors shaping catering system diners’ food choices. Their personal views of contextual factors, such as convenience and fitting for diners’ needs, the existent routine, and the discussion of responsibility for healthy eating in the dining room, were significant factors of food choices when eating in a catering system.

Diners’ freedom of choice was found in this study as a central factor regarding food choices. The foods offered in catering systems are predetermined, often without consulting diners. Promoting healthy dietary choices may entail changes of and in the dishes served. These changes may infringe on diners’ freedom of choice. Thus, planning policies aimed at influencing the choice of healthier foods by diners in a catering system must consider diners’ wish for freedom of choice.

This study echoes the knowledge regarding the factors that shape food choices and extend them to diners in catering systems. Our findings may improve catering healthy nutrition policy planning, as well as its adjustment for diners by communicating with them.

Further research is needed to identify diners’ personal and contextual factors influencing food choices in varied catering systems. Implementation of different food choices modalities and their ongoing changes also deserves further study.

## Figures and Tables

**Table 1 ijerph-19-01885-t001:** The study’s interview guide.

	Main Issue	Questions
1	Healthy eating	What do” health” and “healthy eating” mean to you?
2	Food choices	How do you choose what to eat in general? What did you choose to eat in the last lunch you had in the dining room? What do you think about your choices? Did you have any hesitation about your choices? If so, please describe them.
3	Responsibility for diners’ healthy eating	Who do you think is responsible for healthy food choices? Please explain why.
4	Effective intervention strategies	What do you think about strategies such as serving only healthy foods or labeling food? How would you feel if these strategies were implemented in the dining room? Would that change your food choices? What would you expect from such an intervention program?

**Table 2 ijerph-19-01885-t002:** Characteristics of the interviewees.

Participant #	Kibbutz No.	Age	Gender	Number of Years Eating in the Dining Room	Number of Times Per Week Eating Lunch in the Dining Room
#1	1	22	F	16	5
#2	1	65	F	45	5
#3	1	57	F	38	4
#4	1	54	F	30	5
#5	1	80	M	60	4
#6	1	29	M	23	6
#7	2	44	M	23	5
#8	2	64	F	40	3
#9	2	35	M	4	5
#10	3	63	F	55	4
#11	3	76	F	55	5
#12	3	82	M	64	5
#13	2	37	M	8	5

**Table 3 ijerph-19-01885-t003:** The thematic framework.

Category	Theme	Sub-Theme		Sub-Sub Theme	
The personal context	Eating as a task	Experience-based eating			
		Eating and emotions			
		Eating to satisfy physical needs			
		Eating according to the diner’s values	Freedom of choice		
			Healthy eating as a value	Perception of what is healthy/unhealthy food	Perception of healthy food as not tasty
	Attempts to control one’s eating	Weight loss diet			
		Morality—Guilt—Remorse			
The contextual aspects of eating in the catering system	Eating in the dining room as a default	Pros: Convenience, fits diner’s needs Cons: Not suitable for all diners			
	Characteristics of the food served	Availability Diversity Appearance Flavor Freshness Health and nutritional components Price			
	Routine	Weekdays versus weekends			
	Personal vs. Public	The responsibility for healthy eating			
		Freedom of choice			

## Data Availability

The corresponding author (O.K.-S.) can provide all original data for review.

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
