# Peer review of "Factors Involved in the Food Choices of Diners in a Kibbutz Communal Dining Room Buffet: A Qualitative Study"

_ijerph, 2022, doi:10.3390/ijerph19031885_

Round 1

Reviewer 1 Report

The manuscript received for review investigates factors that are involved in diners’ food choices in a communal dining setting throughout qualitative study, where the topic is relevant for the journal scope and to the present public health problems.

Manuscript contains enough significant original material and it is clearly and concisely written.

The title of the manuscript is informative and corresponds to the content of the work.

The manuscript is provided with a detailed introduction that fully reflect the topic that is investigated.

The goal of the study should be more precisely defined.

The Materials and Methods section describes recruitment, data collection and analysis and ethical approval, all necessary information that allows detail investigation.

The results section provides personal context and contextual aspects. Discussion section elaborates, explains and supports obtained results.

Conclusion section needs some broadening.

Some minor corrections are noted in manuscripts’ pdf file.

Reviewer recommendation: Minor revision.

Author Response

Manuscript ID: ijerph-1550277

Factors Involved in the Food Choices of Diners in a Communal Dining Room Buffet: A Qualitative Study.

January 28, 2022

Responses to Reviewer #1:                                                                                                           We would like to thank the reviewer for his/her helpful comments. Our responses are listed below. For convenience, we numbered the reviewer’s comments.

  1. The manuscript received for review investigates factors that are involved in diners’ food choices in a communal dining setting throughout qualitative study, where the topic is relevant for the journal scope and to the present public health problems.

Manuscript contains enough significant original material and it is clearly and concisely written.

The title of the manuscript is informative and corresponds to the content of the work.

The manuscript is provided with a detailed introduction that fully reflect the topic that is investigated.

Response: Thank you very much for this positive feedback.

  1. The goal of the study should be more precisely defined.

Response: Thank you for your attention. We have reviewed the text following this comment. We rephrased the description of the goal. It now reads (lines 69-74): "The aim of this study was to utilize this unique setting to provide a more nuanced understanding of the factors involved in the food choices of regular diners in a kibbutz communal catering buffet. To achieve this goal, we adopted a qualitative approach, for the examination of the perceptions of the diners about food, their food choices and healthy nutrition in the context of eating in the kibbutz dining room."

  1. The Materials and Methods section describes recruitment, data collection and analysis and ethical approval, all necessary information that allows detail investigation.

The results section provides personal context and contextual aspects. Discussion section elaborates, explains and supports obtained results.

Response: Thank you very much for this positive feedback.

  1. Conclusion section needs some broadening.

Response: Thank you. Following the reviewer’s comment, we revised the Conclusion section. It now reads (lines 587-609): "This work identified the factors involved in the food choices of those who eat regularly in a catering system in a kibbutz. The two categories of themes found dealt with the personal context and the contextual aspects of eating in the catering system. In the personal context category, experience, emotions, physical needs, values, and perceptions were found as factors of eating as a task and of attempts to control one’s eating. In the context of eating in the catering system category, food-related internal and external characteristics were found as factors shaping catering system diners' food choices. Their personal views of contextual factors such as convenience and fitting for diners’ needs, the existent routine, and the discussion of responsibility for healthy eating in the dining room, were significant factors of food choices when eating in a catering system.

Diners’ freedom of choice was found in this study as a central factor regarding food choices. The foods offered in catering systems are predetermined, often without consulting diners. Promoting healthy dietary choices may entail changes of and in the dishes served. These changes may infringe on diners’ freedom of choice. Thus, planning policies aimed at influencing the choice of healthier foods by diners in a catering system, must consider diners’ wish for freedom of choice.

This study echoes the knowledge regarding the factors that shape food choices and extend them to diners in catering systems. Our findings may improve catering healthy nutrition policy planning, as well as its adjustment for diners by communicating with them.

Further research is needed to identify diners’ personal and contextual factors influencing food choices in varied catering systems. Implementation of different food choices modalities and their ongoing changes also deserves further study."

  1. Some minor corrections are noted in manuscripts’ pdf file.
  2. Provide more precise goal of this research

Response: Thank you. Please see our response to comment #2.

  1. This research is 7.5 years old. Provide the explanation for the time period from the research until publishing the results. Has something changed in the meantime?

Response: Indeed, the study was conducted in 2014 as part of the NEKST research project. After completing its qualitative component, the focus shifted to the development of the intervention program (based on the qualitative component's results), the implementation of the intervention, data collection and analyses of results that were all time and resource consuming. Thus, although this study had a crucial part in designing the consecutive parts of the study, its drafting as a scientific report was materialized only recently. We should note however, that in the years since, there haven't been dramatic changes in the operation and dining in the kibbutzim dining rooms.

  1. Line 92-96: Excessive information, try to condense the paragraph.

Response: Thank you. Following the reviewer’s comment, we revised this section. It now reads (lines 108-109): "The interviews took place considering each participant convenience. "

  1. Line 552-574: Rewrite this section, so the conclusion section is broadened.

Response: Please see our response to comment #4.

Reviewer 2 Report

I can make 3 comments:
- the research comes from 2014 and may be partially outdated, but is an excellent basis for developing the topic,
- there is a lack of a broader context of life in a kibbutz related to home cooking, access to commercial products and sources of learning about healthy food,
- I did not find a recommendation for a canteen regarding communication with customers, but it can be developed in subsequent studies.

Besides, I can see many advantages of this article:
- despite the niche community, it shows an interesting nutritional perspective,
- the research is abundantly and interestingly described,
- I can see the involvement of researchers in the process, but also the self-evaluation of the results,
- research should be expanded to include new aspects, such as the flexitarianism or veg trend.

In my PDF file I see strange font in literature numbers and incorrect formatting in literature.
I will be happy to come back to this article after it is posted.

Author Response

Manuscript ID: ijerph-1550277

Factors Involved in the Food Choices of Diners in a Communal Dining Room Buffet: A Qualitative Study.

January 28, 2022

Responses to Reviewer #2:                                                                                                           We would like to thank the reviewer for his/her helpful comments. Our responses are listed below. For convenience, we numbered the reviewer’s comments.

  1. the research comes from 2014 and may be partially outdated, but is an excellent basis for developing the topic.

Response: Thank you very much for the positive feedback. Indeed, the study was conducted in 2014 as part of the NEKST research project. After completing its qualitative component, the focus shifted to the development of the intervention program (based on the qualitative component's results), the implementation of the intervention, data collection and analyses of results that were all time and resource consuming. Thus, although this study had a crucial part in designing the consecutive parts of the study, its drafting as a scientific report was materialized only recently. We should note however, that in the years since, there haven't been dramatic changes in the operation and dining in the kibbutzim dining rooms.

  1. there is a lack of a broader context of life in a kibbutz related to home cooking, access to commercial products and sources of learning about healthy food.

Response: Thank you for your comment. We revised the text to address it. It now reads (lines 54-56):"Kibbutz members and their families may have their meals together, yet they can also choose whether and how to use this communal facility. Alternatively, they can eat at home using food products available in the market."

  1. I did not find a recommendation for a canteen regarding communication with customers, but it can be developed in subsequent studies.

Response: Thank you for your comment. We revised the text and now address this. Please see lines 604-606. "Our findings may improve catering healthy nutrition policy planning, as well as its adjustment for diners by communicating with them."

  1. Besides, I can see many advantages of this article:
    -despite the niche community, it shows an interesting nutritional perspective,
    - the research is abundantly and interestingly described,
    - I can see the involvement of researchers in the process, but also the self-evaluation of the results.

Response: Thank you very much for this comment.

  1. research should be expanded to include new aspects, such as the flexitarianism or veg trend.

Response: Thank you for your comment. We revised the text and now address this. Please see lines 608-609. "Implementation of different food choices modalities and their ongoing changes also deserves further study."

  1. In my PDF file I see strange font in literature numbers and incorrect formatting in literature.
    I will be happy to come back to this article after it is posted.

Response: Thank you. It is now corrected.

Reviewer 3 Report

The paper “Factors Involved in the Food Choices of Diners in a Communal Dining Room Buffet: A Qualitative Study” contributes to the growth of literature for nutritionists as well as food producers offering products for selected kibbutzim consumers.

Before the manuscript acceptation for publication in “Int. J. Environ. Res. Public Health” the following items should be revised:

The analysis is based on a sample of selected consumers of Communal Dining Room Buffet - from three kibbutzim located in the south of Israel.  

That is why I suggest including the subject, purpose and conclusions associated with this: kibbutzim, region and country.

Introduction

After Line 53

what are the positive and negative aspects of nutrition in a kibbutz - nutritional value

Table 2. this is the characteristics of the interviewees - I suggest adding table 2 to The methods.

Discussion

Line 515 – 518 the sentences: “Regarding the contextual aspects of eating in the catering system, one theme dealt with the issue of whom is responsible for the diners' healthy eating. Food providers may offer unhealthy foods, which are often desired the most ( 20 )” - Whether the research from 20 is concern according to kibbutzim consumers?

Line 562 – 563

Notwithstanding these limitations, this work identified the factors involved in the food choices of those who eat regularly in a catering system.“ – The research was concerned according to the catering system of kibbutzim.

Conclusions

The Conclusions was concerning according to kibbutzim consumers.

Author Response

Manuscript ID: ijerph-1550277

Factors Involved in the Food Choices of Diners in a Communal Dining Room Buffet: A Qualitative Study.

January 28, 2022

Responses to Reviewer #3:                                                                                                           We would like to thank the reviewer for his/her helpful comments. Our responses are listed below. For convenience, we numbered the reviewer’s comments.

The paper “Factors Involved in the Food Choices of Diners in a Communal Dining Room Buffet: A Qualitative Study” contributes to the growth of literature for nutritionists as well as food producers offering products for selected kibbutzim consumers.

Before the manuscript acceptation for publication in “Int. J. Environ. Res. Public Health” the following items should be revised:

  1. The analysis is based on a sample of selected consumers of Communal Dining Room Buffet - from three kibbutzim located in the south of Israel.  

That is why I suggest including the subject, purpose and conclusions associated with this: kibbutzim, region and country.

Response: Indeed, our analysis is based on a sample of selected consumers. Indeed, the kibbutz is unique to Israel, but its communal dining-room is similar to other institutional catering systems, such as workplaces, military bases, hospitals, etc. The specific region was not addressed since no significant differences exist between kibbutzim in different areas in of the country. We revised the title, objective, and conclusions to address this comment. It now reads: "Factors Involved in the Food Choices of Diners in a Kibbutz Communal Dining Room Buffet: A Qualitative Study" (Line 2), "The aim of this study was to utilize this unique setting to provide a more nuanced understanding of the factors involved in the food choices of regular diners in a kibbutz communal catering buffet." (Line 71), "This work identified the factors involved in the food choices of those who eat regularly in a catering system in a kibbutz."(Line 588)

Introduction

  1. After Line 53

what are the positive and negative aspects of nutrition in a kibbutz - nutritional value

Response: Thank you for your comment. We revised the text to address this comment. It now reads (lines 61-66):

"The nutritional benefits of the food in the kibbutz dining room are the accessible, fresh, and diverse cooked food served. On the other hand, no healthy food policy is routinely implemented in these communal dining rooms. In most cases, they do not consult with a dietitian regularly to cook and serve healthy food. Therefore, the dishes served often include extra-processed food products, and the use of food ingredients rich in undesired nutrients such as sugar, salt, saturated fat and trans fats."

  1. Table 2. this is the characteristics of the interviewees - I suggest adding table 2 to The methods.

Response: Thank you for the advice. This table can be put in the methods section as well as in the results section. We leave the decision on the location of Table 2 to the editor.

Discussion

  1. Line 515 – 518 the sentences: “Regarding the contextual aspects of eating in the catering system, one theme dealt with the issue of whom is responsible for the diners' healthy eating. Food providers may offer unhealthy foods, which are often desired the most ( 20 )” - Whether the research from 20 is concern according to kibbutzim consumers?

Response: Reference 20 (now 21) does not specifically address kibbutzim consumers. It is a textbook that deals with the psychology of eating in the context of healthy eating.

  1. Line 562 – 563

Notwithstanding these limitations, this work identified the factors involved in the food choices of those who eat regularly in a catering system.“ – The research was concerned according to the catering system of kibbutzim.

Response: Please see our response to comment #1.

  1. Conclusions
  2. The Conclusions was concerning according to kibbutzim consumers.

Response: Please see our response to comment #1.

Reviewer 4 Report

The manuscript ‘Factors Involved in the Food Choices of Diners in a Communal Dining Room Buffet: A Qualitative Study’ is case a study to identify factors responsible for food choices of diners. The paper is written more like a descriptive story than the scientific paper. The problem presented in the paper is interesting but the manuscript needs serious improvements before publication.

COMMENTS:

Major:

  • All results followed by statistical analysis should be added to supplementary material
  • All questionnaires should be attached as a supplementary information and remove from the main text.
  • Extensive editing of English language and style is required. Authors should provide certificate.
  • The abstract should be completely rewritten. The current shape is hardly understandable for a potential reader. Authors should follow the structure: motivation, goal, methodology, main findings.
  • The conclusion section should be seriously extended. Authors should include there the main findings regarding the previously defined goal – factor that determine the food choices. The justification how their findings can be applied, e.g. how identified factors can improve catering systems or how findings from the study can serve to promote healthy and dietary choices.
  • The introduction should be rewritten. Deeper analysis of the literature is needed and the description of the state of the art in the area of factors that determine food choices in other places, locations…

  • 42 – ‘(plural of kibbutz)’ – should be removed
  • 57-61 – the clear description what is the goal of the study should be included. There is no information what authors want to study: map the factor, understand the factor…
  • 71 – ‘who responded to the ad’ – ‘ad’ should be explained
  • 72 – ‘convenience sampling (9) was followed by criteria sampling (9)’ – authors should describe both methods of sampling. The paper should be selfexplanatory without the need to find other papers
  • 66-77 – the recruitment stage should be extended. The way of choosing participants is not clear and needs deep justification from statistical point of view. The justification to use the number of 13 participants to get any conclusions from statistical point of view should be added as well. What is the role and meaning of the head of the health committee to recruit volunteers? Volunteers should be chosen randomly to be representative for the analysis. The number of participants should be extended.
  • 75-77 – the stop criteria is not described in an understandable way. What is the meaning of the term ‘no new information emerged from the interviews, was reached’ – justification of the connection with attached references is needed
  • Table 1 should be moved to the supplementary material
  • 102 – Data analysis section should be completely rewritten. in ‘Data analysis’ section only the detailed description of the methodologies with references should be presented.
  • Data from table 2 should be summarized in one row.
  • Table 3 should be moved to the supplementary material
  • Section 3.1. should be completely rewritten
  • Section 3.1.1 should be renamed as e.g. social factors of food consumption. The entire section should be rewritten. The story, citations etc. should be replaced by presentation of the filtered results focusing on collected factors from interview. The analysis should focus on factors concluded more from groups of participants than from individuals
  • Section 3.1.2 should be rewritten following comments regarding section 3.1.1
  • Section 3.2. should be completely rewritten. Results and description should be presented in a form that can be analyzed by the reader and justify factors recognized by authors, e.g. section 3.2.1 delivers no information, section 3.2.2 should consist of real characteristics of food served in kibbutz, etc.
  • Section Results should consist of list of identified factors that determine food choices.
  • The discussion section should be extended by comparison of data between authors’ study and other catering systems, comparison of data regarding health and weight control from other study vs authors’ own study, etc.

Author Response

Manuscript ID: ijerph-1550277

Factors Involved in the Food Choices of Diners in a Communal Dining Room Buffet: A Qualitative Study.

January 28, 2022

Responses to Reviewer #4:                                                                                                           We would like to thank the reviewer for his/her helpful comments. Our responses are listed below. For convenience, we have turned the reviewer comments' bullets into numbers.

  1. The paper is written more like a descriptive story than the scientific paper. The problem presented in the paper is interesting but the manuscript needs serious improvements before publication.

Response: This paper is indeed descriptive due to its qualitative nature. Description is a fundamental part of the qualitative methodology, and its use makes this methodology compliant with the expected guidelines of scientific writing.

  1. All results followed by statistical analysis should be added to supplementary material.

Response: The analytical approach of qualitative analysis refrains from using statistical tools. Instead, its scientific insights are gained through the systematic analysis of texts, such as customary in the thematic analysis we used for this research.

  1. All questionnaires should be attached as a supplementary information and remove from the main text.

Response: Table 1, the study’s interview guide, is placed in the text for the readers’ convenience. If the editorial team considers it needs to be presented as supplementary information, we can remove it from the main text.

  1. Extensive editing of English language and style is required. Authors should provide certificate.

Response: An English professional editor proofread the document.

  1. The abstract should be completely rewritten. The current shape is hardly understandable for a potential reader. Authors should follow the structure: motivation, goal, methodology, main findings.

Response: The abstract is structured following the reporting practices of qualitative research. In them, our motivation is expressed as " Eating in a catering system may provide an environment for promoting healthy dietary choices." Our goal is stated as " To map the factors that shape the food choices of diners who routinely eat in catering systems". The study methodology is described as " we conducted in-depth, semi-structured, face-to-face interviews with 13 diners, who ate at least three lunches per week in the kibbutz’s dining room. Data analysis followed thematic analysis principles ". The main findings appear in lines 17-22. The study conclusion is written in lines 22-24.

  1. The conclusion section should be seriously extended. Authors should include there the main findings regarding the previously defined goal – factor that determine the food choices. The justification how their findings can be applied, e.g. how identified factors can improve catering systems or how findings from the study can serve to promote healthy and dietary choices.

Response: Thank you very much for your comment. The conclusion section was extensively reviewed and enriched (lines 587-609). It now reads: "This work identified the factors involved in the food choices of those who eat regularly in a catering system in a kibbutz. The two categories of themes found dealt with the personal context and the contextual aspects of eating in the catering system. In the personal context category, experience, emotions, physical needs, values, and perceptions were found as factors of eating as a task and of attempts to control one’s eating. In the context of eating in the catering system category, food-related internal and external characteristics were found as factors shaping catering system diners' food choices. Their personal views of contextual factors such as convenience and fitting for diners’ needs, the existent routine, and the discussion of responsibility for healthy eating in the dining room, were significant factors of food choices when eating in a catering system.

Diners’ freedom of choice was found in this study as a central factor regarding food choices. The foods offered in catering systems are predetermined, often without consulting diners. Promoting healthy dietary choices may entail changes of and in the dishes served. These changes may infringe on diners’ freedom of choice. Thus, planning policies aimed at influencing the choice of healthier foods by diners in a catering system, must consider diners’ wish for freedom of choice.

This study echoes the knowledge regarding the factors that shape food choices and extend them to diners in catering systems. Our findings may improve catering healthy nutrition policy planning, as well as its adjustment for diners by communicating with them.

Further research is needed to identify diners’ personal and contextual factors influencing food choices in varied catering systems. Implementation of different food choices modalities and their ongoing changes also deserves further study."

  1. The introduction should be rewritten. Deeper analysis of the literature is needed and the description of the state of the art in the area of factors that determine food choices in other places, locations…

Response: Thank you for your comment. Following your comment we expanded the introduction. Please see lines 34-38. "The factors known as influencing food choices include food-related internal factor (sensory and perceptual features), food-related external (environmental) factors, personal factors (physiological and psychological characteristics, habits and experiences), cognitive factors (knowledge, attitude and personal identity) as well as sociocultural factors(6)."

  1. Line 42 – ‘(plural of kibbutz)’ – should be removed.

Response: Done.

  1. Line 57-61 – the clear description what is the goal of the study should be included. There is no information what authors want to study: map the factor, understand the factor…

Response: Thank you very much for your comment. We rephrased the description of the goal. It now reads (lines 69-74): "The aim of this study was to utilize this unique setting to provide a more nuanced understanding of the factors involved in the food choices of regular diners in a kibbutz communal catering buffet. To achieve this goal, we adopted a qualitative approach, for the examination of the perceptions of the diners about food, their food choices and healthy nutrition in the context of eating in the kibbutz dining room."

  1. Line 71 – ‘who responded to the ad’ – ‘ad’ should be explained

Response: Thank you for your comment. It now reads (lines 81-86):"Members of the three kibbutzim were asked via an advertisement to voluntarily participate in an interview. The advertisement was sent by email to the kibbutz community and posted on bulletin boards in the dining rooms. The advertisement requested the participation of regular diners who ate at least three lunches a week in the communal dining room. First, we recruited 10 kibbutz members who responded to the advertisement."

  1. Line 72 – ‘convenience sampling (9) was followed by criteria sampling (9)’ – authors should describe both methods of sampling. The paper should be selfexplanatory without the need to find other papers

Response: Participants’ recruitment followed the practices of qualitative research. The convenience sampling is detailed in lines 81-85 (please, see our answer to comment #10) and the criteria sampling is detailed in lines 87-90, as follows: " …was followed by criteria sampling(10) in which we reached out to participants for whom healthy nutrition was a less important value. The head of the health committee of each kibbutz, who served as the contact person for the study, facilitated this step. In this second stage, we recruited three participants. Interviews were conducted until theoretical saturation, meaning no new information emerged from the interviews, was reached(11)."

.

  1. Line 66-77 – the recruitment stage should be extended. The way of choosing participants is not clear and needs deep justification from statistical point of view. The justification to use the number of 13 participants to get any conclusions from statistical point of view should be added as well. What is the role and meaning of the head of the health committee to recruit volunteers? Volunteers should be chosen randomly to be representative for the analysis. The number of participants should be extended.

Response: As customary in qualitative studies, a statistically representative sample is not applicable. Instead, we used a purposeful sample in which each participant was carefully selected for the specific purposes of the research. In view of the characteristics of the first 10 participants, we looked for those who could potentially complement the emerging themes. Thus, and as customary with qualitative research sampling strategies, we reached out for a person familiar with potential participants, specifically requiring participants whose views were not yet reflected in our sampling. Naturally, the 10 first participants were persons with a higher interest in health issues, thus their agreement to participate in our research. Since we also wanted to include the views of persons less health-oriented, the head of the health committee was instrumental in helping to reach out to persons answering to this recruiting criteria.

  1. Line 75-77 – the stop criteria is not described in an understandable way. What is the meaning of the term ‘no new information emerged from the interviews, was reached’ – justification of the connection with attached references is needed

Response: “Theoretical saturation” is a commonly used term in qualitative research, which means that no new ideas emerge from the collection of more data. This is explained in the text: "Interviews were conducted until theoretical saturation, meaning no new information emerged from the interviews, was reached."(lines 90-92).

  1. Table 1 should be moved to the supplementary material

Response: Table 1, the study’s interview guide, is placed in the text for the readers’ convenience. If the editorial team considers it needs to be presented as supplementary information, we can remove it from the main text.

  1. Line 102 – Data analysis section should be completely rewritten. in ‘Data analysis’ section only the detailed description of the methodologies with references should be presented.

Response: The written in the data analysis section followed the practices of qualitative research reporting. Lines 119-124describe the steps of the thematic analysis implemented. Lines 124-130 describe the steps taken to ensure study trustworthiness (the equivalent of reliability in qualitative methodology). Lines 130-136 describe the steps taken to retain the original meanings after the translation from Hebrew to English.

  1. Data from table 2 should be summarized in one row.

Response: By offering the details about each participant’s characteristics, this table allows the readership to contextualize the quoted texts. This is why the participant number appears in each quote.

  1. Table 3 should be moved to the supplementary material

Response: This table shows the core results of the study. It presents and summarizes the main themes and sub-themes derived from the interviews. It also allows the reader to graphically see the "big issues" and their specific components.

  1. Section 3.1. should be completely rewritten

Section 3.1.1 should be renamed as e.g. social factors of food consumption. The entire section should be rewritten. The story, citations etc. should be replaced by presentation of the filtered results focusing on collected factors from interview. The analysis should focus on factors concluded more from groups of participants than from individuals

Response: Qualitative data analysis seeks each participant’s nuanced, contextualized, specific view, rather than a group generalization. Its power in this meticulous description is further reflected in the punctilious selection of the titles. This is achieved through iterative readings of the texts, and the discussion of all researchers, in order to arrive at what better reflects the expressed by the participants. The identification of themes is not based on the statistical frequency of the words, but on the ideas behind them.

  1. Section 3.1.2 should be rewritten following comments regarding section 3.1.1

Response: Please see our response to comment #18.

  1. Section 3.2. should be completely rewritten. Results and description should be presented in a form that can be analyzed by the reader and justify factors recognized by authors, e.g. section 3.2.1 delivers no information, section 3.2.2 should consist of real characteristics of food served in kibbutz, etc.

Response: Please see our response to comment #18.

The characteristics of the food are detailed as sub-themes in table 3. All were pointed by the participants (Availability, Diversity, Appearance, Flavor, Freshness, Health and nutritional components, Price).

  1. Section Results should consist of list of identified factors that determine food choices.

Response: Please see Table 3 for the list of identified factors that determine food choices.

  1. The discussion section should be extended by comparison of data between authors’ study and other catering systems, comparison of data regarding health and weight control from other study vs authors’ own study, etc.

Response: Indeed, the discussion deals with factors that have emerged as shaping diners’ food choice and what is known from the literature about the effect of these factors. We also discuss the contextual aspects of eating in a catering system, including its pros and cons. The discussion also compares our findings to other studies' findings. In addition, the discussion emphasizes the finding that freedom of choice is significant for the diners, and its possible implications.

Round 2

Reviewer 4 Report

Author addressed all comments included in review form.